# Gut Microbiota and Its Repercussion in Parkinson's Disease: A Systematic Review in Occidental Patients

Ana Cristina Proano [1,*], Javier A. Viteri [2], Eileen N. Orozco [3], Marco Antonio Calle [4], Stefany Carolina Costa [3], Daniel V. Reyes [1], Melissa German-Montenegro [3], David F. Moncayo [5], Andrea C. Tobar [3] and Juan A. Moncayo [6]

1 School of Medicine, Universidad Internacional del Ecuador, Quito 170411, Ecuador; danielreyescev@gmail.com
2 School of Medicine, Colegio de Ciencias de la Salud, Universidad San Francisco de Quito, Cumbayá 170901, Ecuador; jav014.vit@gmail.com
3 School of Medicine, Pontificia Universidad Católica del Ecuador, Quito 170143, Ecuador; guigu_eom@hotmail.com (E.N.O.); stefycarolinacr@gmail.com (S.C.C.); meli.g.m.13@gmail.com (M.G.-M.); mdandreatobar1294@live.com (A.C.T.)
4 School of Medicine, Universidad Católica de Santiago de Guayaquil, Guayaquil 090615, Ecuador; marcocalle.md@gmail.com
5 Department of Gastroenterology, Pontificia Universidad Católica del Ecuador, Quito 170143, Ecuador; davidmonc@hotmail.com
6 Department of Neurology, Texas Tech University Health Sciences Center, Lubbock, TX 79439, USA; jmoncayo725@gmail.com
* Correspondence: acristinapro@gmail.com

**Abstract:** (1) Background: Parkinson's disease (PD) is a relatively common and complex pathology, and some of its mechanisms remain to be elucidated. Change in host microbiota is related to the pathophysiology of numerous diseases. This systematic review aims to gather existing data on the occidental hemisphere, compare it, and search for any significant association between Parkinson's disease and gut microbiota dysbiosis. (2) Methods: Preferred Reporting Items for Systematic Review and Meta-Analysis (PRISMA) and Meta-analyses Of Observational Studies in Epidemiology (MOOSE) protocols were used for this systematic review. PubMed was used as the database search engine. Of the 166 studies found, only 10 were used, as they met our inclusion criteria: case–control studies, studies that assessed the correlation of PD and gut microbiome, studies that took place in occidental regions, and studies that were performed on humans and were written in English. The Newcastle–Ottawa Scale was used as the assessment tool for overall risk of bias in this systematic review. (3) Results: The studies analyzed were divided into three geographic areas: Region 1: United States of America and Canada; Region 2: Germany, Ireland, and Finland; and Region 3: Italy; based on geographical similarities among these populations. The following statistically significant results were described in PD patients, compared with non-PD controls. In the first region, a significant increase in the following bacteria was seen: 1. Phylum: Actinobacteriota and its Genus: *Bifidobacterium*; 2. Phylum: Verrucomicrobiota and its Genus: *Akkermansia*; 3. Genus: *Enterococcus*, *Hungatella*, *Lactobacillus*, and *Oscillospira* of the Phylum: Firmicutes; 4. Family: *Ruminococcaceae* of Phylum: Firmicutes; 5. Phylum: Bacteroidetes and its Genus: *Bacteroides*; 6. Phylum: Proteobacteria. A significant decrease was described in the Family: *Lachnospiraceae* and its Genus: *Blautia*, *Coprococcus*, and *Roseburia*, which belong to the Phylum: Firmicutes. In the second region, a raised number of: 1. Phylum: Verrucomicrobiota, its Genus: *Akkermansia*, and its Species: *Akkermansia muciniphila*; 2. Family: *Verrucomicrobiaceae* of the Phylum: Verrucomicrobiota; 3. Genus: *Lactobacillus* and *Roseburia* of the Phylum: Firmicutes; 4. Family: *Lactobacillaceae* of the Phylum: Firmicutes; 5. Family: *Barnesiellaceae* of the Phylum: Bacteroidetes; 6. Genus: *Bifidobacterium* of the Phylum: Actinobacteriota; 7. Species: *Bilophila wadsworthia* of the Phylum: Thermodesulfobacteriota, was identified. Only one Genus: *Prevotella* of the Phylum: Bacteroidetes was decreased. In the third and last region, an augmented number of these bacteria were found: 1. Phylum: Verrucomicrobiota and its Genus: *Akkermansia*; 2. Family: *Bifidobacteriaceae* and *Coriobacteriaceae* of the Phylum: Actinobacteriota; 3. Phylum: Firmicutes and its Family: *Christensenellaceae* and *Lactobacillaceae*; 4. Family: *Enterococcaceae*

and its Genus: *Enterococcus*, of the Phylum: Firmicutes; 5. Genus: *Lactococcus* and *Oscillospira*, of the Phylum: Firmicutes; 6. Phylum: Proteobacteria, its Family: *Enterobacteriaceae*, and the Genus: *Citrobacter*, *Klebsiella*, *Salmonella*, and *Shigella*; 7. Genus: *ParaBacteroides* of the Phylum: Bacteroidetes. In contrast, a significant decrease in 1. Phylum: Firmicutes, its Family: *Lachnospiraceae*, and its Genus: *Roseburia* and 2. Genus: *Ruminococcus* of the Phylum: Firmicutes, was described. (4) Conclusion: A significant gut dysbiosis, involving multiple bacterial taxa, was found in PD patients compared to healthy people in the occidental regions. However, more studies are needed to find the precise pathophysiologic involvement of other groups of pathogens, such as fungi and parasites, in the development and progression of PD.

**Keywords:** Parkinson's disease; gut; microbiota; microbiome; dysbiosis; short-chain fatty acids; occidental regions; Brain-Gut Axis

## 1. Introduction

The gut microbiome is composed of microorganisms such as bacteria, viruses, and yeasts. Each individual has a unique microbiota that has developed since birth and continues molding within the first years of life [1]. Due to several environmental factors, it will thrive and achieve equilibrium during a lifetime. The microbiota plays an essential role in the symbiosis of the host, from digestion and absorption of nutrients to the protection against pathogens and education of the immune system [2,3].

Due to its influence on physiological processes, it has been proposed that any imbalance in this environment can have negative consequences in the host, leading to a dysregulation in pro-inflammatory and anti-inflammatory factors, causing disorders in distinct locations other than the gut, such as the central nervous system (CNS) [3]. The term "dysbiosis" refers to the disturbance of the bacterial species that make up the microbiota, and it has been the target of several studies in recent years.

It is well-known that the CNS has a selective immune defense driven by the blood–brain barrier (BBB). The BBB consists of the endothelial cells from the blood vessels that supply blood to the brain. These cells are interconnected with tight junctions, allowing a rigorous passage of cells, electrolytes, and nutrients from the systemic circulation into the brain [4]. However, other particles, such as metabolites (amino acids, essential vitamins) and bacterial products, can cross the BBB; this whole interaction forms the so-called Gut–Brain Axis (GBA). This complex is a two-way communication system that influences the immune, endocrine, and enteric nervous systems [5,6].

Due to this communication and early exposure, Socala et al. propose that the pathogenesis of certain neurological conditions has a root in this complex system, such as Alzheimer's disease, Parkinson's disease, multiple sclerosis, depression, and anxiety [7].

Parkinson's disease (PD) is a chronic, neurodegenerative disorder characterized by motor symptoms, starting from mild tremors to difficulty walking due to stiffness; these motor symptoms are the Parkinsonian triad: rigidity, bradykinesia, and tremor. PD is also associated with other nonmotor symptoms such as cognitive impairment, depression, psychosis, sleep disturbances, and gastrointestinal disturbances in more than 80% of patients [8]. Gastrointestinal symptoms such as constipation, gastroparesis, and weight loss appear in the early stages of the disease [9], even before the development of motor symptoms. The pathogenesis of PD occurs due to damage to dopaminergic neurons in the substantia nigra pars compacta. The pathologic accumulation of Lewy bodies (mainly α-synuclein) in the presynaptic neurons is responsible for neuronal cell death [7], which causes the broad spectrum of manifestations in PD as represented in Figure 1.

The gut microbiota is one of the proposed mechanisms of signaling and inflammation in the CNS; these molecular changes induce a systemic response and further depositing of amyloid in the presynaptic space. Bacterial fermentation from dietary nutrients produces short-chain fatty acids (SCFAs); during absorption, some molecules can travel

from the blood vessels and cross the BBB [10]. Some of these SCFAs are precursors of neurotransmitters such as GABA, serotonin, glutamate, adrenaline, noradrenaline, and dopamine [11]—the latter being a cornerstone in the pathophysiology of PD. However, some specific SCFAs, such as butyrate kinase, are produced by bacteria. These products impact the down-regulation of inflammation [12], and their absence could lead to a pro-inflammatory state.

There is increasing evidence for the involvement of neuroinflammatory molecules in the pathogenesis of PD [13]. Bacterial metabolites, such as Endotoxin, can trigger a local and systemic inflammatory response, increasing TNF-a, IL-8, and IL-6 levels, leading to dysregulated activation of glial cells and causing pathological α-Synuclein accumulation [14]. Li et al. found that the use of naturally occurring intraluminal antibiotics (some found in traditional Chinese medicine) can increase blood/fecal levodopa levels through changes in the intestinal flora, with an improvement of motor and cognitive symptoms, avoiding the need to increase standard treatment doses per os; they also found that an upgrade of the microbiome to a noninflammatory type led to a significant reduction in TNF-α, IL-6, and IL-8 serum levels, which could stop microglia activation and, therefore, theoretically stop substantia nigra degeneration [15].

Regardless of new evidence of these associations, and due to PD multifactorial pathogenesis, our team decided to further study this perplexing communication between the gut and the brain. In this systematic review, we aim to provide a thorough analysis of the impact of gut microbiota on the pathophysiology of PD.

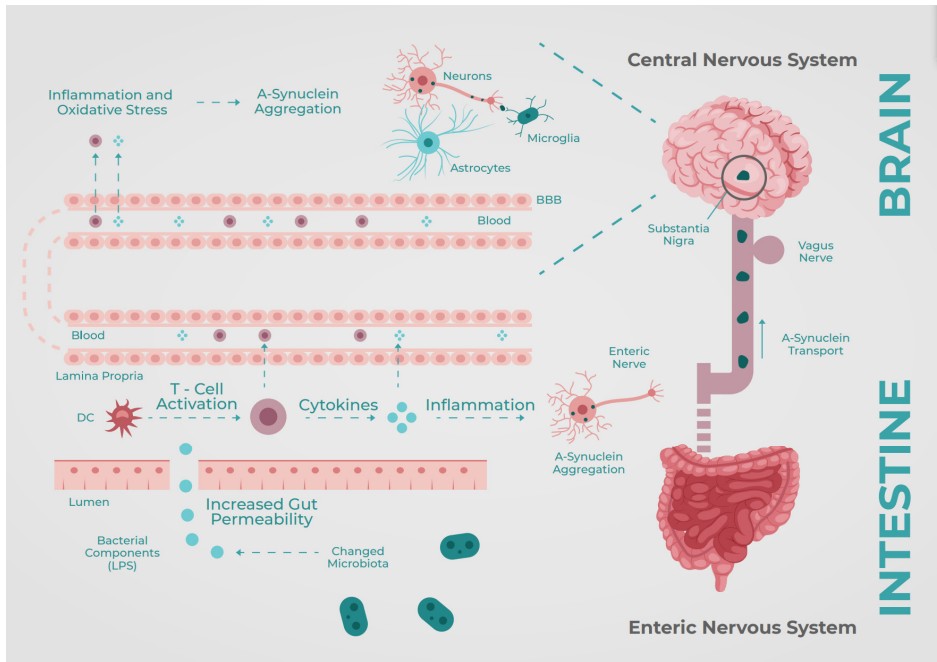

**Figure 1.** A-synuclein Brain–Gut Axis. BBB: Blood–brain barrier, DC: Dendritic Cells, LPS: Lypopolysaccharides. Based on Perez-Pardo et al. [16].

## 2. Materials and Methods

### 2.1. Protocol

We carried out a systematic review using the PRISMA and MOOSE protocols (Figure 2). This systematic review was developed and reported following the Preferred Reporting Items for Systematic Review and Meta-analysis (PRISMA) guidelines which are evidence-based and consist of a minimum set of items focusing on the reporting of reviews evaluating various types of research. Before the formal screening of search results, the protocol for this study was registered in PROSPERO 2023 (link: https://www.crd.york.ac.uk/

 registration number CRD42023421796.

## 2.2. Eligibility Criteria and Study Selection

Inclusion criteria were clinical trials conducted on humans and written in English. Exclusion criteria included animal studies, studies that did not assess PD and gut microbiome, and articles that did not fulfill the aim of our research. As described in Figure 2, we only included articles about PD and its association with gut microbiome. After applying these filters, we discarded 156 studies and accepted 10 studies.

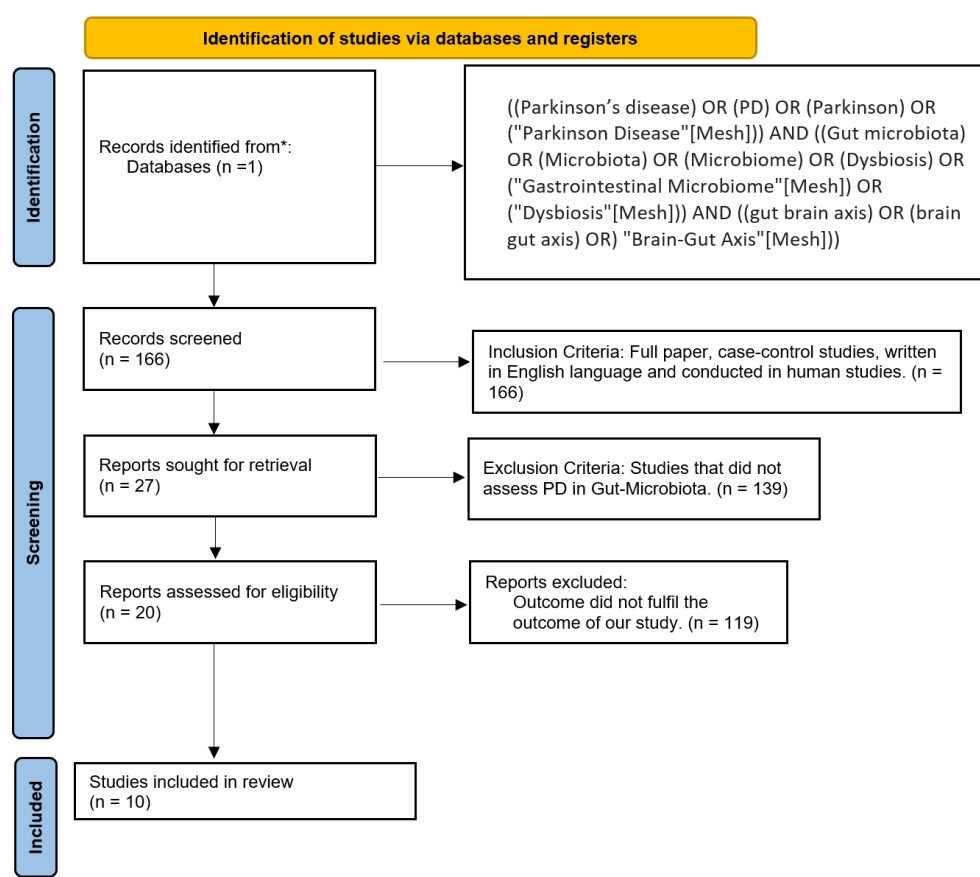

**Figure 2.** PRISMA flow chart. * PubMed.

The purpose of this study is to review current evidence on the different species involved, since the microbiome is related to the sociodemographic characteristics of patients, and to address if there is a shared pathway despite these variables. We divided the studies into three groups because diet and habits are among the crucial factors influencing the microbiome. They differentiate qualitatively among the regions included but could have some similarities related to disease progression.

## 2.3. Database and Search Strategy

We used the PubMed database for this systematic meta-analysis review. We extrapolated the data between 12 April and 1 May 2023. We used an advanced search strategy with the following terms: "((Parkinson's disease) OR (PD) OR (Parkinson) OR ("Parkinson Disease" [Mesh])) AND ((Gut microbiota) OR (Microbiota) OR (Microbiome) OR (Dysbiosis) OR ("Gastrointestinal Microbiome" [Mesh]) OR ("Dysbiosis" [Mesh])) AND ((gut brain axis) OR (brain gut axis) OR ("Brain-Gut Axis" [Mesh]))".

*2.4. Data Extraction and Analysis*

From each paper we collected: author/year/country, methods, number of participants, and study design. We also extracted the main results, including the outcome measures and limitations of each observational/clinical trial. We analyzed the studies' primary and secondary goals and the main conclusions from each study for further analysis divided into 3 subgroups.

Upon selection of the articles, we grouped them into three categories based on their region: Region 1: USA and Canada, Region 2: Germany, Ireland, and Finland, and Region 3: Italy, as described in Figure 3.

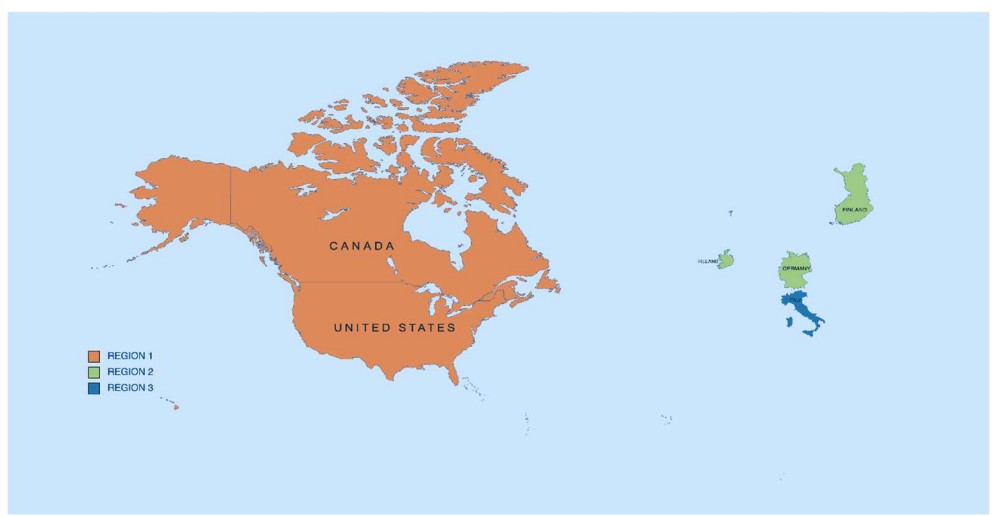

**Figure 3.** Map: Region 1 (orange), Region 2 (green), and Region 3 (blue) [16].

*2.5. Bias Analysis*

The assessment of overall risk for bias in this systematic review was conducted with the Newcastle–Ottawa Scale, which divides the risk into three categories: high, moderate, and low; depending on its score, from 0 to 3, 4 to 6, and 7 to 9, respectively. With this tool, the systematic review has a moderate risk of overall bias, as demonstrated in Table 1.

**Table 1.** Newcastle–Ottawa Scale.

| Study | Newcastle–Ottawa Scale | | | Overall Risk of Bias |
|---|---|---|---|---|
| | Selection (Total 4) | Comparability (Total 2) | Outcome/Exposure (Total 3) | (Total score) |
| Keshavarzian et al., 2015, USA [17]. | *** | ** | * | Moderate (6) |
| Hill-Burns et al., 2017, USA [18]. | ** | ** | ** | Moderate (6) |
| Appel-Cresswell et al., 2020, Canada [19]. | - | * | * | High (2) |
| Zhang et al., 2020, USA [20]. | **** | ** | *** | Low (9) |
| Pietrucci et al., 2019, Italy [21]. | *** | * | *** | Moderate (6) |
| Cereda et al., 2018, Italy [22]. | *** | * | ** | Moderate (6) |
| Hopfner et al., 2017, Germany [23]. | **** | * | ** | Low (7) |
| Heintz-Buschart et al., 2017, Germany [24]. | *** | ** | ** | Low (7) |
| Hertel, J. et al., 2019, Ireland [25]. | *** | ** | ** | Low (7) |
| Aho et al., 2019, Finland [26]. | *** | * | *** | Moderate (6) |

Score: 0–3: High Risk for Bias, 4–6: Moderate Risk for Bias, 7–9: Low Risk for Bias. * Good Quality: 3 or 4 stars in selection domain AND 1 or 2 stars in comparability domain AND 3 or 4 stars in outcome/exposure domain. Fair Quality: 2 stars in selection domain AND 1 or 2 stars in comparability domain AND 2 or 3 stars in outcome/exposure domain. Poor Quality: 0 or 1 stars in selection domain OR 0 stars in comparability domain OR 0 or 1 in outcome/exposure domain.

## 3. Results

### 3.1. Region 1

Within Region 1 (Figure 3; Tables 2 and 3), we found four articles. Keshavarzian et al. [17] (USA) investigated 38 PD patients and compared them to 34 controls. Their study showed an increase in Proteobacteria, Verrucomicrobiota, Bacteroidetes, *Akkermansia*, *Oscillospira*, and *Bacteroides*, and decreased *Lachnospiraceae*, *Blautia*, *Coprococcus*, and *Roseburia* in fecal samples. The authors mentioned that the characteristics between groups in terms of age, body mass index, previous medications, and duration of the disease were not considered and could have affected the results.

On the other hand, Hill-Burns et al. [18] (USA) accepted 327 patients divided into 197 cases and 130 controls. The fecal specimens showed a reduction in *Lachnospiraceae*, with an increase in *Akkermansia*, *Lactobacillus*, and *Bifidobacterium* compared to the controls. Variables such as PD medication, disease duration, and geographic location were cataloged as confounders and recognized as limitations for their study.

*Lachnospiraceae* was found to be reduced by Keshavarzian et al. [17] (USA) and Hill-Burns et al. [18] (USA), which is related to butyrate kinase production, an anti-inflammatory molecule; therefore, with a lower number of these bacteria, a pro-inflammatory setting would be present.

In the rural areas of California, Zhang et al. [20] (USA) recruited 170 patients, with 96 cases and 74 controls. The authors found that patients with PD had an increased biome of Proteobacteria, Verrucomicrobiota, Actinobacteriota, UBA1819 (*Ruminococcaceae*), DTU089 (*Ruminococcaceae*), *Akkermansia*, *Enterococcus*, and *Hungatella*. A compelling aspect of this research is that they classified the cases into motor-symptom subtypes (Postural Instability and Gait dysfunction [PIGD], Tremor Dominant [TD], and Indeterminate [IND]) and noticed that the PIGD motor subtype was more abundant in Verrucomicrobiota. In contrast, the other subtypes highlighted the presence of Proteobacteria. Their theory proposes that elevated lipopolysaccharides (LPS) in Gram-negative bacteria's walls, such as Proteobacteria, Verrucomicrobiota, and *Akkermansia*, trigger the immune system. As data collection was at a single time, they stated that neither temporality nor causality can be determined. Additionally, they specified that their sample size, confounders control, and statistical power represented the limitations of this research.

The uniqueness between Hill-Burns et al. [18] (USA) and Zhang et al. [20] (USA) among their selection groups is that they used healthy controls (HC) living in the same household, and thus were comparable.

Most studies research bacteria; however, Appel-Cresswell et al. [19] (Canada) focused theirs on fungi. Through microbial DNA sequencing, 152 fecal samples were tested and contrasted with 95 cases versus 57 controls. The most common was Saccharomyces in 94% of all participants, followed by *Candida* spp. with 35%, and Cladosporium and Penicillium in last place with 23%.

### 3.2. Region 2

In Region 2 (Figure 2; Tables 4 and 5), we selected four studies performed in countries on the northern side of Europe. Hertel et al. [25] (Ireland) investigated 60 subjects from the German longitudinal de novo Parkinson's disease cohort; patients received dopaminergic modulators. The authors chose 30 PD patients and compared them to 30 healthy control volunteers. This investigation determined a rise of *Akkermansia muciniphila* and *Bilophila wadsworthia* within the cases. These two bacteria are associated with sulfur metabolism, whose byproducts are known to be related to local inflammation and neurotoxicity in the brain. Unlike the rest of the studies, they used the Virtual Metabolic Database to recognize bacteria and metabolites. Additionally, habits such as diet and exercise were considered, which could affect the conclusions.

**Table 2.** Region 1—USA and Canada.

| Author/Country/Year * | Sample Number | Methodology | Microbiota Alteration in PD | Mechanisms (Theories) | Association | Limitations |
|---|---|---|---|---|---|---|
| Keshavarzian et al. [17], 2015, USA | Cases: (n = 38) Controls: (n = 34) | Fecal and colonic mucosa samples: Microbial 16S rRNA gene on genomic DNA. | - Less abundant in fecal samples: Family: *Lachnospiraceae* Genus: *Blautia*, *Coprococcus*, *Roseburia* - Less abundant in mucosa samples: Family: Coprobacillaceae Genus: Dorea, *Faecalibacterium* - More abundant in fecal samples: Phylum: Bacteroidetes, Proteobacteria, Verrucomicrobiota Genus: *Akkermansia*, *Oscillospira*, *Bacteroides* - More abundant in mucosa samples: Family: Oxalobacteraceae Genus: Ralstonia. | Family *Lachnospiraceae* contains several butyrate-producing bacteria, butyrate has anti-inflammatory properties. Thus, the low abundance of butyrate-producing bacteria in feces from PD subjects may be one mechanism contributing to intestinal leakiness and inflammation in PD. | Yes | Environmental factors associated with PD (age, BMI, medication use, and PD duration) might explain differences between cases and control. The study found significant differences in age between PD and controls, but its impact is uncertain. |
| Hill-Burns et al. [18], 2017, USA | Cases: (n = 197) Controls PD-household: (n = 130) | Fecal samples: 16S rRNA amplicon sequencing on genomic DNA. | -Less abundant in fecal samples: Family: *Lachnospiraceae* - More abundant in fecal samples: Genus: *Akkermansia*, *Lactobacillus*, *Bifidobacterium*. | SCFA are made by bacteria in the gut; notably, *Lachnospiraceae* is consistent with its depletion. Butyrate kinase, which catalyzes a reversible reaction between butyrate and butanoylphosphate, was reduced in PD, and acetyl-CoA synthetase, which converts acetate to acetyl-CoA, was elevated. | Yes | Most relevant potential confounders: PD medications, disease duration, marital status, and geographic site. |
| Appel-Cresswell et al. [19], 2020, Canada | Cases: (n = 95) Controls: (n = 57) | Fecal samples: Microbial DNA was extracted using QIAGEN and amplicon base sequenced. | -Mycobiome found in fecal samples: Phylum: 86% Ascomycota,12% Basidiomycota Genus: Saccharomyces (most common 58.7%) in 94% of participants, Candida 35%, Cladosporium 23%, Penicillium 23%. | Found no correlation between PD patients and control fecal samples. | No | - Only one ITS primer pair was used for sequencing. - Additional data on environmental factors were not collected. |
| Zhang et al. [20], 2022, USA | Cases: (n = 96) Controls PD-household: (n = 74) | Fecal samples: 16S rRNA gene sequencing, rarefaction, and feature filtering. | - More abundant in fecal samples: Phylum: Proteobacteria, Verrucomicrobiota, and Actinobacteriota. Genus: UBA1819 (*Ruminococcaceae*), DTU089 (*Ruminococcaceae*), *Akkermansia*, *Enterococcus*, and *Hungatella*. | Increased level of Proteobacteria has been associated with potential immunoregulation ability via the production of LPS. They initiate the immune process, stimulating microglia and leading to dopaminergic neuron necrosis. | Yes | - Gastrointestinal comorbidities. - The recent use of antibiotics was not specified. - Size, confounders control, and statistical power. - The samples were collected at a single time; this does not allow for establishing temporality or causality. - The species-level resolution of the sequencing and annotation pipeline. |

* Case–control studies; cases were PD patients. PD: Parkinson's disease; SCFA: short-chain fatty acids; GI: gastrointestinal; ITS2: Internal transcribed spacer 2; ITS: Internal transcribed spacer; LPS: lipopolysaccharides.

**Table 3.** Microbiota changes of PD patients VS controls in Region 1.

| | # | Genus | Family | Order | Class | Phylum |
|---|---|---|---|---|---|---|
| **Actinobacteriota** | ↑ | | | | | ✔ |
| *Akkermansia* | ↑ | ✔ | *Akkermansia*ceae | Verrucomicrobiales | Verrucomicrobiae | Verrucomicrobiota |
| *Bacteroides* | ↑ | ✔ | Bacteroidaceae | Bacteroidales | Bacteroidia | Bacteroidetes |
| **Bacteroidetes** | ↑ | | | | | ✔ |
| *Bifidobacterium* | ↑ | ✔ | *Bifidobacteriaceae* | Bifidobacteriales | Actinomycetia | Actinobacteriota |
| *Blautia* | ↓ | ✔ | *Lachnospiraceae* | Eubacteriales | Clostridia | Firmicutes |
| *Coprococcus* | ↓ | ✔ | *Lachnospiraceae* | Eubacteriales | Clostridia | Firmicutes |
| *Enterococcus* | ↑ | ✔ | *Enterococcaceae* | Lactobacillales | Bacilli | Firmicutes |
| *Hungatella* | ↑ | ✔ | *Clostridiaceae* | Clostridiales | Clostridia | Firmicutes |
| *Lachnospiraceae* | ↓ | | ✔ | Eubacteriales | Clostridia | Firmicutes |
| *Lactobacillus* | ↑ | ✔ | *Lactobacillaceae* | Lactobacillales | Bacilli | Firmicutes |
| *Oscillospira* | ↑ | ✔ | *Oscillospira*ceae | Eubacteriales | Clostridia | Firmicutes |
| **Proteobacteria** | ↑ | | | | | ✔ |
| *Roseburia* | ↓ | ✔ | *Lachnospiraceae* | Eubacteriales | Clostridia | Firmicutes |
| *Ruminococcaceae* | ↑ | | ✔ | Clostridiales | Clostridia | Firmicutes |
| **Verrucomicrobiota** | ↑ | | | | | ✔ |

Visual representation of bacteria found in Region 1 with their complete taxonomic classification for better understanding. #: quantity of bacteria in PD patients compared to controls; ↑: increased; ↓: decreased; ✔: bacterial taxonomic rank.

Aho et al. [26] (Finland) conducted a study with 128 subjects divided into 64 patients and 64 healthy participants. The bacteria they found most abundant in the case section were *Bifidobacterium*, *Lactobacillus*, and *Roseburia*. A secondary outcome of this research was the result of decreased levels of *Prevotella* in patients with a faster progression of the disease. This last finding led the researchers to conclude that reducing *Prevotella* in the intestinal lumen could influence the correlation to PD. The writers stated further investigation should be conducted with patients taking similar medication as PD patients (e.g., restless legs syndrome or de novo patients starting medication) as these drugs could influence the microbiome.

We selected two studies from Germany. Hopfner et al. [23] recruited 29 PD cases and 29 healthy volunteers. This investigation showed an increment in *Lactobacillaceae* and *Barnesiellaceae*. They did not state any possible mechanism through which these bacteria could influence the pathogenesis of this condition. The lack of information regarding diet habits, related factors, and cardiovascular comorbidities in the HC group raises the suspicion of possible confounders.

Heintz-Buschart et al. [24], with 99 patients and 76 controls, observed an increase in *Akkermansia* and *Verrucomicrobiaceae* in the samples of PD patients. The authors agreed with the current hypotheses that *Akkermansia* spp. is critical in the secretion of α-synuclein and future deposit in the CNS. Similar to Hopfner et al. [23], in this study diabetes and coronary artery disease were comorbidities present among the healthy participants, and the oral intake of medication could modify the microbiome.

In addition to the phenomenon in which bacteria could reach the CNS from the olfactory bulb, Heinz-Buschart et al. [24] have also included 76 nasal wash samples from PD patients and 78 HCs. However, the study showed no significant variation between both groups, showing that pathologic pathways may differ from the nasal and gut microbiota in PD patients. Nonetheless, the only significant difference was found in the Bacillaceae family, in a group of PD patients treated with L-dopa, suggesting that its variation might be attributed to the medication rather than the disease.

**Table 4.** Region 2—Germany, Ireland, and Finland.

| Author/Country/Year * | Sample Number | Methodology | Microbiota Alteration in PD | Mechanisms (Theories) | Association | Limitations |
|---|---|---|---|---|---|---|
| Hopfner et al., 2017, Germany [23]. | Cases: (n = 29) Controls: (n = 29) | Samples: Next-generation sequencing of the 16S rRNA gene. | - Significant increase in PD patients of Family: *Lactobacillaceae* - Higher abundance in PD cases of: Family: *Barnesiellaceae*. | The GI microbiome influences the enteric nervous system and through the vagal nerve reaches the central nervous system. | Yes | - Previous dietary habits, also cardiovascular comorbidities were present in the control group. - Cases were using PD treatment. |
| Heintz-buschart et al., 2017, Germany [24]. ** | Cases: (n = 99) Controls: (n = 76) | Samples: 16S and 18S ribosomal RNA amplicon sequencing from flash-frozen stool and nasal swab. | - Relative abundances in the PD patients of: Genus: *Akkermansia* Family: *Verrucomicrobiaceae* - In addition, 75% showed the same changes in the gut microbiome between the PD patients and RBD vs. the HC: Genus: Anaerotruncus, Clostridium XIVb. Phylum: Bacteroidetes. | *Akkermansia* spp. abundance is related to higher susceptibility to a pathogen due to the depletion of the mucus layer. | Yes | - Comorbidity of diabetes and coronary artery disease were found to affect the prokaryotic taxonomic profiles. - Oral intake of diabetes medication was considered a potential confounder. |
| Hertel et al., 2019, Ireland [25]. | Cases: (n = 30) Controls: (n = 30) | - Targeted metabolomic analysis. - Computational analysis methods and the Virtual Metabolic Human database. | - Significant increase in abundance of: Genus: *Akkermansia muciniphila* and *Bilophila wadsworthia*. | A. muciniphila produces hydrogen sulfide, which is pro-inflammatory and harmful to the gut. This correlates to gastrointestinal motility dysfunction and higher absorption of bacterial toxins through the gut barrier in PD patients. B. Wadsworthia produces sulfite, which is pro-inflammatory to the gut and a neurotoxin. | Yes | - The effects of exercise were not monitored. - The study results could be influenced by dietary variance. - The study results cannot describe the mechanics involved in PD presentation and metabolic alterations. |
| Aho et al., 2019, Finland [26]. | Cases: (n = 64) Controls: (n = 64) | Stool samples: twice on average 2–2.5 years apart: 16s rRNA gene amplicon sequencing. | - Increased in fecal samples in PD patients at baseline and follow-up: Genus: *Bifidobacterium*, *Lactobacillus*, *Roseburia*—progressed PD patients had a Firmicutes-dominated enterotype. - Less abundant in PD patients and faster progression patients: Genus: *Prevotella*. | The early nonmotor symptoms of PD led to the hypothesis that it could originate outside the CNS, in the enteric nervous system. | Yes | - Deficient follow-up. - PD intake medication stroke or TIA was common among the controls. |

* Case–control studies; cases were PD patients. ** Cases included PD patients (n = 78) and RBD patients (n = 21). PD: Parkinson's disease; GI: gastrointestinal; RBD: REM sleep behavior disorder; HC: healthy controls; CNS: central nervous system; TIA: transient ischemic attack.

**Table 5.** Microbiota changes of PD patients VS controls in Region 2.

| | # | Spp | Genus | Family | Order | Class | Phylum |
|---|---|---|---|---|---|---|---|
| *Akkermansia* | ↑ | | ✔ | *Akkermansia*ceae | Verrucomicrobiales | Verrucomicrobiae | Verrucomicrobiota |
| *Akkermansia Muciniphila* | ↑ | ✔ | *Akkermansia* | *Akkermansia*ceae | Verrucomicrobiales | Verrucomicrobiae | Verrucomicrobiota |
| *Barnesiellaceae* | ↑ | | | ✔ | Bacteroidales | Bacteroidia | Bacteroidetes |
| *Bifidobacterium* | ↑ | | ✔ | *Bifidobacteriaceae* | Bifidobacteriales | Actinomycetia | Actinobacteriota |
| *Bilophila wadsworthia* | ↑ | ✔ | Bilophila | Desulfovibrionaceae | Desulfovibrionales | Desulfovibrionia | Thermodesulfobacteriota |
| *Lactobacillaceae* | ↑ | | | ✔ | Lactobacillales | Bacilli | Firmicutes |
| *Lactobacillus* | ↑ | | ✔ | *Lactobacillaceae* | Lactobacillales | Bacilli | Firmicutes |
| *Prevotella* | ↓ | | ✔ | *Prevotella*ceae | Bacteroidales | Bacteroidia | Bacteroidetes |
| *Roseburia* | ↑ | | ✔ | *Lachnospiraceae* | Eubacteriales | Clostridia | Firmicutes |
| *Verrucomicrobiaceae* | ↑ | | | ✔ | Verrucomicrobiales | Verrucomicrobiae | Verrucomicrobiota |
| *Verrucomicrobiota* | ↑ | | | | | | ✔ |

Visual representation of bacteria found in Region 2 with their complete taxonomic classification for better understanding. #: quantity of bacteria measured compared to controls; ↑: increased; ↓: decreased; SP: species; ✔: bacterial taxonomic rank.

### 3.3. Region 3

In Region 3 (Figure 3; Tables 6 and 7), we grouped two studies made in Italy mainly due to their specific Mediterranean diet. Piertrucci et al. [21] gathered 152 patients; the case group (N = 80) showed an elevation in the levels of Proteobacteria, *Lactobacillaceae* (phylum: Firmicutes), *Enterobacteriaceae*, *Enterococcaceae*, *Citrobacter*, *Enterococcus*, *Lactococcus*, *Klebsiella*, *Salmonella*, and *Shigella*, and a reduction in *Lachnospiraceae* (phylum: Firmicutes) and *Roseburia* in their fecal samples when compared to the control group (N = 72). The accumulation of LPS in the intestine was postulated again as a possible mechanism of inflammatory response. Additionally, this research found that the levels of phenylalanine, tryptophan, and tyrosine decreased in PD patients; nonetheless, they did not postulate any specific reason for this finding. Based on the premise that catechol-O-methyltransferase inhibitors (COMTi) alter the microbiota, eight patients taking this medication were extracted from the case group. They consisted of part of a subgroup and exhibited no change in the presence of *Citrobacter*, *Enterococcus*, *Lactococcus*, *Klebsiella*, *Salmonella*, *Shigella*, an unclassified *Enterobacteriaceae*, and *Roseburia* as to non-PD participants. There was a significant difference in age, sex, and weight loss between the two groups; these variables could be considered potential confounders.

After collecting a cohort of 193 PD patients and 113 controls, Cereda et al.'s [22] research yielded an increase in *Akkermansia*, Proteobacteria, *Enterobacteriaceae*, *Christensenellaceae*, *Lactobacillaceae*, *Coriobacteriaceae*, *Bifidobacteriaceae*, *ParaBacteroides*, *Oscillospira*, and Verrucomicrobiota and a decrease in *Lachnospiraceae*, *Roseburia*, and Rominococcus in PD cases. As previously mentioned, *Lachnospiraceae* is associated with butyrate-related SCFA production; their metabolites preserve the integrity of the local immune system, hence the intestinal barrier. Moreover, the authors mentioned that *Lactobacillaceae* was elevated in the case group, and plays a role in α-synuclein secretion, a hallmark in the pathogenesis of PD. The writers tried to match the controls to the cases; however, the lack of biopsies from the sigmoid colon and metagenomic analyses were labeled as possible limitations.

**Table 6.** Region 3—Italy.

| Author/Country/Year * | Sample Number | Methodology | Microbiota Alteration in PD | Mechanisms (Theories) | Association | Limitations |
|---|---|---|---|---|---|---|
| Pietrucci et al., 2019, Italy [21]. | Cases: (n = 80) Controls: (n = 72) | Fecal samples: 16S rRNA amplicon (V3–V4 regions) sequencing analysis PD-COMTi takers: 8. | - Higher levels in the feces compared to controls: Phylum: Proteobacteria, Firmicutes Family: *Enterobacteriaceae, Lactobacillaceae, Enterococcaceae* <br> - Lower levels in the feces compared to controls: Phylum: Firmicutes Family: *Lachnospiraceae* <br> - More abundant in PD- COMTi: Genus: *Citrobacter, Enterococcus, Lactococcus, Klebsiella, Salmonella, Shigella*, and an unclassified *Enterobacteriaceae* <br> - Less abundant in PD- COMTi: Genus: *Roseburia*. | Gut microorganisms metabolize phenylalanine, tryptophan, and tyrosine. This study found they were reduced in PD patients. An increase in the presence of Gram-negative bacteria leads to the elevation of LPS content. Thus, increasing intestinal permeability. | Yes | - Unbalanced cases and controls. <br> - Age and sex. <br> - Loss of 5 kg in the last year. |
| Cereda et al., 2018, Italy [22]. | Cases: (n = 193) Controls: (n = 113) | Fecal samples: 16S gene ribosomal RNA sequencing. | - Higher levels in fecal samples of PD compared to controls: Phylum: Verrucomicrobiota, Proteobacteria Family: *Enterobacteriaceae, Lactobacillaceae, Coriobacteriaceae, Bifidobacteriaceae* Genus: *Akkermansia, Christensenellaceae, ParaBacteroides, Oscillospira* <br> - Lower levels in fecal samples of PD compared to controls: Family: *Lachnospiraceae* Genus: *Roseburia, Ruminococcus*. | Production of SCFA such as butyrate, propionate, and acetate. Additionally, *Lactobacillaceae* is involved in the activity of the enteric nervous system and its secretion of α-synuclein and brain chemistry through the vagal nerve. | Yes | - Lack of sigmoid mucosal biopsies. <br> - Lack of characterization of microbial functions through metagenomic analyses. <br> - Presence of irritable bowel syndrome. |

* Case–control studies; cases were PD patients. PD: Parkinson's disease; COMTi: catechol-O-methyltransferase inhibitors; LPS: lipopolysaccharides.

**Table 7.** Microbiota changes of PD patients VS controls in Region 3.

| | # | Genus | Family | Order | Class | Phylum |
|---|---|---|---|---|---|---|
| *Akkermansia* | ↑ | ✔ | *Akkermansia*ceae | Verrucomicrobiales | Verrucomicrobiae | Verrucomicrobiota |
| *Bifidobacteriaceae* | ↑ | | ✔ | Bifidobacteriales | Actinomycetia | Actinobacteriota |
| *Christensenellaceae* | ↑ | | ✔ | Eubacteriales | Clostridia | Firmicutes |
| *Citrobacter* | ↑ | ✔ | *Enterobacteriaceae* | Enterobacterales | Gammaproteobacteria | Proteobacteria |
| *Coriobacteriaceae* | ↑ | | ✔ | Coriobacteriales | Coriobacteriia | Actinobacteriota |
| *Enterobacteriaceae* | ↑ | | ✔ | Enterobacterales | Gammaproteobacteria | Proteobacteria |
| *Enterococcaceae* | ↑ | | ✔ | Lactobacillales | Bacilli | Firmicutes |
| *Enterococcus* | ↑ | ✔ | *Enterococcaceae* | Lactobacillales | Bacilli | Firmicutes |
| Firmicutes * | ↑↓ | | | | | ✔ |
| *Klebsiella* | ↑ | ✔ | *Enterobacteriaceae* | Enterobacterales | Gammaproteobacteria | Proteobacteria |
| *Lachnospiraceae* | ↓ | | ✔ | Eubacteriales | Clostridia | Firmicutes |
| *Lactobacillaceae* | ↑ | | ✔ | Lactobacillales | Bacilli | Firmicutes |
| *Lactococcus* | ↑ | ✔ | Streptococcaceae | Lactobacillales | Bacilli | Firmicutes |
| *Oscillospira* | ↑ | ✔ | *Oscillospira*ceae | Eubacteriales | Clostridia | Firmicutes |
| *ParaBacteroides* | ↑ | ✔ | Tannerellaceae | Bacteroidales | Bacteroidia | Bacteroidetes |
| Proteobacteria | ↑ | | | | | ✔ |
| *Roseburia* | ↓ | ✔ | *Lachnospiraceae* | Eubacteriales | Clostridia | Firmicutes |
| *Ruminococcus* | ↓ | ✔ | *Oscillospira*ceae | Eubacteriales | Clostridia | Firmicutes |
| *Salmonella* | ↑ | ✔ | *Enterobacteriaceae* | Enterobacterales | Gammaproteobacteria | Proteobacteria |
| *Shigella* | ↑ | ✔ | *Enterobacteriaceae* | Enterobacterales | Gammaproteobacteria | Proteobacteria |
| **Verrucomicrobiota** | ↑ | | | | | ✔ |

Visual representation of bacteria found in Region 3 with their complete taxonomic classification for better understanding. #: quantity of bacteria in PD patients compared to controls; ↑: increased; ↓: decreased; ✔: bacterial taxonomic rank. * Both *Lachnospiraceae* (decreased) and *Lactobacillaceae* (increased) belong to the phylum Firmicutes.

## 4. Discussion

Parkinson's disease is one of the most common neurodegenerative disorders. Its complex pathogenesis includes mitochondrial dysfunction, oxidative stress, intracellular protein accumulation, and abnormal protein degradation. Dysbiosis has been described in PD patients, but whether this happens as a cause or a consequence of disease progression remains unclear. There is growing evidence that the microbiome could be one of the most crucial factors contributing to disease progression and its clinical manifestations. Additionally, nonmotor symptoms (mainly gastrointestinal dysfunction) often precede by decades before their motor manifestations and diagnosis.

Our review attempts to address PD from a first principle thinking perspective, aiming at the true causes behind the pathophysiology of this disease and not only its clinical manifestations. Current treatments focus on the effects of substantia nigra degeneration, and a way to properly halt disease appearance and its progression remains elusive.

This systemic review suggests changes in the colonic microbiota in patients with Parkinson's disease in the three regions of our study.

There are alterations in the microbiota of patients with Parkinson's disease; some bacteria are more common than others, depending on their geographic region. Both USA and Italy regions have an increase in Proteobacteria and a decrease in *Lachnospiraceae* in their fecal samples, while in the northern side of Europe *Lactobacillaceae* and *Akkermansia* are the most prevalent; in the case of Finland, they reported a reduction in *Prevotella*. This may be mainly due to the similarities in the diet (probably Mediterranean/European) of the last two regions; however, other confounders could influence the result, such as genetics, medication utilization, and lifestyle. In Germany, the nasal microbiota was analyzed in patients with PD, since it is proposed that the toxin from the bacteria can reach the CNS from the olfactory bulb, which is also affected in patients with PD due to the aggregation of the $\alpha$-synuclein. Therefore, hyposmia is also characteristic in these patients. However, there was no significant difference in the nasal microbiota.

Since the decrease in butyrate-related SCFA production and the increase in LPS are hypotheses that need to be confirmed, it is of the utmost importance to further investigate the mechanisms by which bacteria influence the pathology of Parkinson's disease.

The limitations of this review were the limited number of case–control studies and the lack of data in some areas, such as Latin America. The long disease progression makes it harder to properly design cohort studies and clinical trials, especially in low-income countries. Diet is usually self-reported, increasing the chance of recall bias.

## 5. Conclusions

A clear association was found between PD and gut dysbiosis when comparing patients in the occidental region, with more similarities between European countries and the Americas. Due to its insidious onset regarding nonmotor symptoms, the screening and caption of PD patients for clinical trials are limited. Latin America is a relegated area in terms of research.

**Author Contributions:** Conceptualization, A.C.P. and J.A.M.; methodology, J.A.M.; validation, D.V.R., A.C.T. and M.G.-M.; formal analysis, M.A.C., S.C.C. and D.F.M.; investigation, A.C.P., S.C.C., A.C.T., J.A.V., E.N.O. and M.A.C.; resources, D.V.R.; writing—original draft preparation, A.C.P., J.A.M. and M.G.-M.; writing—review and editing, S.C.C., J.A.V. and E.N.O.; supervision, J.A.M.; project administration, A.C.P. All authors have read and agreed to the published version of the manuscript.

**Funding:** This research received no external funding.

**Institutional Review Board Statement:** Not applicable.

**Informed Consent Statement:** Not applicable.

**Data Availability Statement:** Not applicable.

**Conflicts of Interest:** The authors declare no conflict of interest.

## Abbreviations

| | |
|---|---|
| BBB | Blood–brain barrier |
| CNS | Central nervous system |
| COMTi | Catechol-O-methyltransferase inhibitors |
| GBA | Gut–Brain Axis |
| HC | Healthy controls |
| IL-6 | Interleukin 6 |
| IL-8 | Interleukin 8 |
| IND | Indeterminate |
| LPS | Lipopolysaccharides |
| MOOSE | Meta-analysis of observational studies in epidemiology |
| PD | Parkinson's disease |
| PIGD | Postural Instability and Gait dysfunction |
| PRISMA | Preferred Reporting Items for Systematic Reviews and Meta-Analyses |
| SCFAs | Short-chain fatty acids |

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
