# Peer review of "Gut Microbiota and Its Repercussion in Parkinson’s Disease: A Systematic Review in Occidental Patients"

_2035-8377, doi:10.3390/neurolint15020047_

Round 1
Reviewer 1 Report
In the. Introduction next to gut microbiota also the oral and in particular the nasal microbiota should be mentioned even if no strong differences were found in the studies available so far.
In the Results section line ... 327 patients divided into 197 cases and 130 controls...
1. What is the main question addressed by the research? Gut brain axis still of Interest concerning the pathophysiology of PD. 2. Do you consider the topic original or relevant in the field? Described already in a number of earlier papers. Of special interest the nasal and oral brain axis as biomarker and potential therapeutic option. 3. What does it add to the subject area compared with other published material? Harmful nasal and oral bacteria possibly involved in the generation of alpha-synuclein. Obviously the loss of smell preceeds PD motor signs in many cases for years / decades. The olfactory mucosa can be easily collected and can also be repeated. 4. What specific improvements should the authors consider regarding the methodology? None. What further controls should be considered? None. 5. Are the conclusions consistent with the evidence and arguments presented and do they address the main question posed? Yes. 6. Are the references appropriate? Yes. 7. Please include any additional comments on the tables and figures. Tables and Figures clear. An additional illustration of the association between the gut, oral and nasal microbiota and PD should be added.
Author Response
Greetings dear reviewer,
As suggested by you we made the following changes and highlighted in the new document:
-New figure (number 1) with a description of the pathways proposed as pathophysiology.
-Discussion was expanded with a new paragraph about A-synuclein and its implication.
Reviewer 2 Report
This review article describes the association between Gut-microbiota and Parkinson’s Disease (PD). This review gathers the data from existing studies, which are mainly from the occidental hemisphere. The authors explain the results of different microbiota species obtained from PD patients in a very thorough manner. There are lots of limitations among those studies, which are thoroughly addressed by the authors. Though this review does not provide a clear picture of microbiota species in PD, it is still very useful for people who have been trying to find the link between microbiota and PD. This review is very well written, and I recommend its publication with some minor changes.
- Tables 2, 4 and 6 have columns for the case-control “study” and PD patients “model”, which do not add anything to the information provided as they are the same in all the studies. Instead, the authors should have added extra columns for sample number, type of methods used, etc. “microbiota alteration in PD” column seems redundant as the same information is provided in much more detail in tables 3, 5, and 7. I recommend the authors organize the tables in a manner, which provides a clear and concise picture to the readers.
English seems perfectly fine except for some minor corrections in spacing between the words and some missing articles.
Author Response
Dear reviewer,
As suggested
The following corrections were made and highlighted in the word document:
- Tables 2, 4, and 6:
- Columns titled “study” and “model” were deleted, and their information was added to the table footnotes.
- A new column titled “sample number” was added and filled with some information previously located in the “methodology” column.
- Tables titles and footnotes were realigned.
- The dimensions of the tables were readjusted.
- The columns titled “Microbiota alteration in PD” were kept, as we think they give the reader a better understanding of each study. These columns denote statistically significant results in the differences of the bacteria found cases vs controls. In contrast, tables 3, 5, and 7 were created because in each study they mention numerous bacteria, and many of them can belong to the same phylum, family, etc. This makes it difficult for a reader, without a deep understanding of bacterial classification, to correctly comprehend the results, as this can be confusing.
- Tables 3, 5, and 7:
- A new footnote was added explaining the purpose of these tables, explaining the difference of them from the “Microbiota alteration in PD” column in tables 2, 4, and 6.
- Bold text was transformed into regular text in the first column of table 7.
- Text:
- Text was rearranged to prevent big blank spaces between the text and the tables, keeping its original chronology.
- In the first paragraph of each region, in the first parenthesis the words “; table #” was added (i.e., Figure 2; Tables 2-3).
- Discussion:
- A new paragraph about nasal microbiota was added taking in consideration the study of Heintz-Buschart. It was described the possible correlation of nasal microbiota and Parkinson’s Disease, being protein alpha synuclein aggregation a potential mechanism but not significant difference was found.